# DCTS: Fusing Discrete and Continuous Information for Time Series Forecasting

## Abstract

In time series analysis, data is usually treated as a set of continuous values. Conventional methods do all the computations, from inputs to outputs, in continuous form. While this continuous representations are highly expressive, it can also pay too much attention to fine-grained details. This risks introducing noise and overlooking critical information. In contrast, discrete representations can assign a single code to each temporal pattern. In this way, key patterns underlain in the data are extracted more effectively, while some informative details are probably filtered out along with noises. In order to combine the advantages of both, we propose fusing discrete and continuous information for time series forecasting (DCTS), that incorporates both continuous and discrete approaches, it fuses the expressive power of continuous encoding and pattern-abstracting ability of discrete encoding. It uses a codebook learned via vector quantization to extract discrete encoding from the time series and then fuses it with the continuous encoding. In doing so, the model can benefit from the strengths of both continuous and discrete representations. Additionally, we use multiple codebooks to encode the time series. A single code can hardly cover the entire feature space of a time series. In contrast, multiple discrete values can be combined, exponentially expanding the encoding space and achieving much stronger expressive power. We evaluated our proposed method on multiple real-world datasets and achieved the best performance compared to the baseline methods.

## 1 Introduction

Time series data is prevalent across various fields(Jin et al., 2024), including weather(Bi et al., 2023), transportation(Li et al., 2023), energy(Jiao et al., 2021), and the environment(Liang et al., 2023). With the rapid advancements in internet and sensor technologies in recent years, time series analysis techniques have become crucial in many domains. As time series data describes the changes of systems over continuous time, most time series analysis methods focus on the continuous information within the data.

Recent research has focused on time series embedding methods. Initially, information from each time point was used as positional embedding(Vaswani et al., 2017). Later, single time series were segmented into patches, with each patch being embedding(Nie et al., 2022). Another strategy involved treating each variable as a token to encode the entire time series(Liu et al., 2023). These embedding methods have propelled the development of time-series forecasting. However, existing methods are largely limited to continuous features. Previously, these approaches mapped continuous time series data into different dimensional continuous vectors, and do all the computations, from inputs to outputs, in continuous form.

Inspired by pre-trained large models in NLP (Natural Language Processing) and CV (Computer Vision), as language and image data can also be considered continuous, many researchers have attempted to apply techniques from these fields to time series data for analyzing its continuous characteristics. Recently, methods based on Vector Quantization for obtaining discrete e from data have seen widespread application. Vector Quantization(Van Den Oord et al., 2017) method quantizes continuous data processed by an encoder layer using a trainable codebook. This process obtains discrete encoding, such as image backgrounds or skin tones. Compared to traditional methods, this

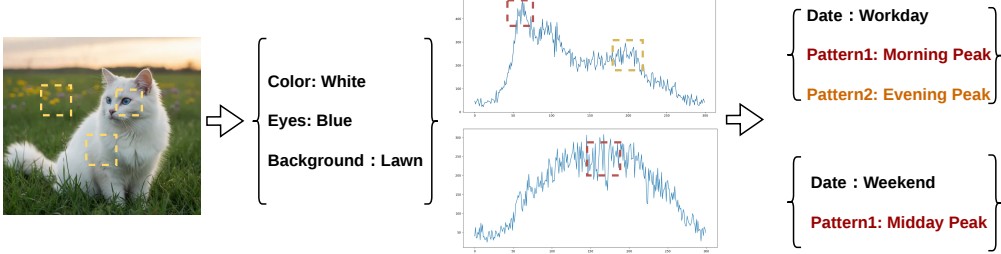

Figure 1: The figure above shows that time series data has clear discrete features, such as different data patterns. This is similar to how the cat in the picture has discrete features like its fur color, eye color, and background.

discrete approach can generate higher-quality images more targeted(Razavi et al., 2019). These methods can also be extended to the field of time series analysis.

Different time series on different days often show distinct patterns in time series data. For instance, as illustrated in Figure 1, traffic volume on weekdays increases significantly around 8 AM and 5 PM, which are peak commute hours. In contrast, on weekends, travel demand is more evenly spread out and not clustered in the morning and evening. Discrete encoding is well-suited for learning these different modes of behavior. Alternatively, traditional continuous encoding methods are effective at capturing the continuous nature of a time series, including fine-grained changes, and they are computationally efficient. Their drawback, however, includes boundary and numerical errors. A classic example is encoding days of the week: Monday follows Sunday, but a continuous encoding might place them at opposite ends of a scale. This creates a large artificial distance and can imply a flawed numerical relationship (e.g., Sunday > Monday). Since discrete codes represent categories without numerical order, supplementing continuous data with them can prevent these issues.

Currently, there are also some studies that apply vector quantization to time series tasks. For instance, SDFormer(Chen et al., 2024) applies Vector Quantization to time series generation tasks. It uses similarity-driven vector quantization, grouping similar time series into categories stored in a codebook. VQShape(Wen et al., 2024) quantizes abstract shapes and offsets of subsequences for time series classification. However, these methods are limited to the traditional Vector Quantization approach with a single codebook. A one-dimensional encoding space struggles to represent all time series data effectively.

Addressing these limitations and targeting time series prediction tasks, we propose DCTS. By adding discrete encoding to traditional models designed for continuous representation, we enhance the model's expressive power through the fusion of discrete and continuous encoding. First, we encode time series data through an encoder layer and quantize it using a learnable codebook.

Traditional quantization relies on a single codebook, which assigns only one discrete code to an entire time series. Such a one-dimensional code is insufficient to cover the full feature space. To address this, we adopt a multi-codebook method. An analogy can be made: the traditional approach uses a single "letter" to describe a time series, whereas our method combines multiple "letters" to create a "word." Given the same number of quantization vectors, these multiple discrete codes can form thousands of different "words." This results in an exponential expansion of the discrete encoding space, which enhances the model's expressive capabilities. Furthermore, to better suit the prediction task, we use a decoder to interpret the resulting codes. We then integrate this discrete encoding into the embedding layer of a model that processes continuous information, supplementing it with discrete information. Our main contributions are summarized as follows:

- We propose a based on Vector Quantization method that quantizes multivariate time series data using a learnable codebook to obtain its discrete information, which are then incorporated into a continous model. This allows the model to fuse discrete and continuous features, thereby improving its performance.
- Unlike traditional single codebooks, we use multiple codebooks for quantizing, extending the one-dimensional code space to multiple dimensions. This enhances the model's ability to represent data features without increasing the codebook size or vector dimensions.

- We conducted extensive experiments on multiple real-world datasets and achieved state-of-the-art results, demonstrating the effectiveness and validity of our method.

## 2 RELATED WORK

### 2.1 TIME SERIES FORECASTING

Time series forecasting has been a key research topic in both academia and industry for decades. It has a wide range of applications in fields like finance(Ding et al., 2015), weather(Bi et al., 2023), and energy(Jiao et al., 2021). Early research on time series forecasting focused on traditional statistical machine learning methods. These included models like the Autoregressive Integrated Moving Average (ARIMA)(Box & Pierce, 1970), Support Vector Regression (SVR)(Cao & Tay, 2003), Gradient Boosting Decision Tree (GBDT)(Xia & Chen, 2017), and Vector Autoregression (VAR)(Biller & Nelson, 2003). These methods were good at handling non-linear relationships and were very efficient. However, because of their simple structure, they could not accurately model complex variable relationships.

Later, the development of deep learning changed the field completely. This included Recurrent Neural Networks (RNNs)(Rangapuram et al., 2018) and Long Short-Term Memory (LSTM) networks for dynamic time series data(Salinas et al., 2020), as well as the Transformer model, which is excellent at capturing relationships in long sequences(Zhou et al., 2021)(Wu et al., 2021)(Zhou et al., 2022). Compared to traditional machine learning, deep learning can capture deeper, more abstract features in the data. This has significantly improved prediction accuracy and the ability to handle datasets with complex variable relationships.

In recent years, to address the low efficiency of Transformer-based models, some researchers have proposed lighter models. These include models based on Convolutional Neural Networks (CNNs)(Zhao et al., 2017)(Borovykh et al., 2017) and Multilayer Perceptrons (MLPs)(Wang et al., 2024)(Han et al., 2024), which analyze long-term information by using 1D convolutions or by decompose the series into trend and seasonal components. At the same time, to offer a new way of looking at time series, some scholars have introduced frequency-domain analysis(Xu et al., 2023). This method transforms data from the time domain to the frequency domain to analyze the relationships between different frequency components.

### 2.2 TIME SERIES EMBEDDING

In current time series forecasting models, various embedding methods are often used to map raw data into higher-dimensional continuous vector representations. This allows the model to fully understand the data's context. Initially, to solve the problem that the self-attention mechanism lacks sequence order information, sine and cosine functions were used to add positional information to each time point(Vaswani et al., 2017). This created time series embeddings that included sequential order. Some researchers also added calendar timestamp information, such as the day of the week or distinctions between workdays and holidays, to analyze data differences. Later, PatchTST introduced Patch Embedding(Nie et al., 2022), which segments the time series into patches and then embeds them. Each embedding contains information from a nearby time period, so time points are no longer isolated. Each embedding represents a local pattern over a period of time. More recently, iTransformer proposed inverted embedding(Liu et al., 2023). This method treats the entire time series of a variable as a single embedding unit, rather than individual time points or patches. This allows the model to directly learn the relationships between variables. These methods have advanced time series analysis, but they only focus on the continuous information in the time series and ignore the discrete information.

Based on vector quantization(Van Den Oord et al., 2017) learns a codebook to map complex, high-dimensional continuous data into a discrete latent space. In the fields of CV(Razavi et al., 2019) and NLP(Baevski et al., 2019), converting images and audio clips into discrete tokens has been shown to generate more realistic images and audio compared to traditional methods. This idea has also been brought into time series analysis. In time series generation tasks, SDFormer(Chen et al., 2024) uses similarity-driven vector quantization to group similar time series into one category stored in the codebook. In time series classification, VQShape uses a codebook to learn abstract shape and offset

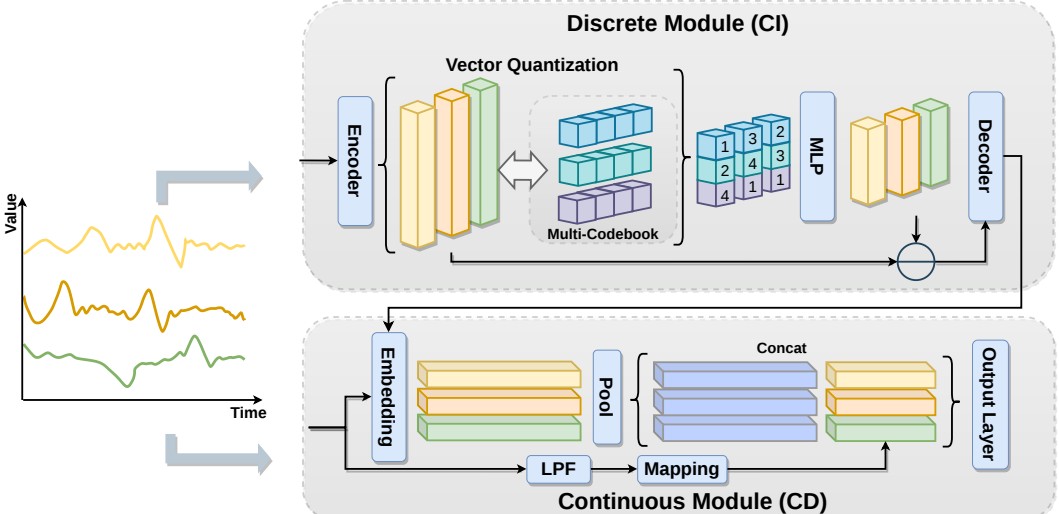

Figure 2: This figure shows the DCTS model architecture, which is comprised of two modules: 1) **Discrete Module** to get discrete information, 2) **Continuous Module** to fuse the continuous and discrete information.

information from the time series(Wen et al., 2024). However, these methods are limited to a single codebook, which results in a small encoding space that struggles to describe the entire feature space.

## 3 METHODOLOGY

We introduce DCTS to overcome the limitations of previous methods. The core idea is to enhance the model's expressive power by incorporating discrete information into a model built for continuous data. The complete model architecture is illustrated in Figure 2. DCTS is composed of two components: 1) the discrete module: responsible for discretely encoding the original data; 2) the continuous module: fuses the discrete information with the continuous information to yield the final forecast.

### 3.1 DISCRETE MODULE

To obtain discrete information for the time series, we introduce a method based on vector quantization. It uses a two-layer convolutional network for both the encoder and decoder. Based on the channel-independent(CI) mechanism, the encoder processes each time series, and its output $z_e(x) \in \mathbb{R}^{N \times d}$ is quantized by a learnable codebook, $N$ is represents the $N$ variables, $d$ is the length of the codebook vector.. Finally, the decoder reconstructs the series from the quantized code. Traditional based on vector quantization typically use a single codebook to quantize a time series, resulting in a single discrete code to describe the data. However, one discrete code is clearly not enough to fully capture the feature space of the entire time series, which limits the model's expressive power.

To address this issue, we employ multiple codebooks to encode the time series, allowing each codebook to learn features independently, $e^i$ represents the $i$-th codebook. This approach significantly expands the discrete encoding space, transforming it from a single dimension to a multi-dimensional one. We measure similarity between the time series and the codes in each codebook using Euclidean distance, consistent with the original vector quantization method. As a result, each codebook independently generates a discrete code $e_k^i$. The collection of these codes $E$ jointly represents the time series. This is then passed through a linear layer to compute a final, weighted discrete representation. following these formulas:

$$z_q(x) = \text{Linear}(E), \; E = \{e_k^1, e_k^2, ...e_k^d\} \quad \text{where} \quad k = \text{argmin}||z_e(x) - e^i||_2 \tag{1}$$

In traditional pre-trained vector quantization models, the decoder is used to reconstruct the original time series from the discrete codes to serve various downstream tasks. However, to make our model more suitable for forecasting, we use the decoder to process the discrete codes. This produces an output containing the discrete information of the time series, which is then integrated into the downstream forecasting task.

### 3.2 CONTINUOUS MODULE

In this section, we design the Continuous Module, which fuses the discrete and continuous information from the time series to generate the final forecast. The process begins by applying the Fourier Transform to convert the time series into the frequency domain. A low-pass filter (LPF) is then used to filter out high-frequency information, which is typically considered noise. Removing this noise smooths the time series, allowing for a focus on more critical patterns. Next, the filtered spectrum is projected to the forecast length by a linear layer and converted back to the time domain. This step yields an initial forecast $\overline{Y}$ ,which captures the fundamental trend and periodic components.

We then merge the raw input with the discrete output from the Discrete Module and feed it into an embedding layer. This process yields a fused representation $\hat{X}$, containing both the discrete and continuous information of each time series. Subsequently, these representations are pooled and fused to create a summary of the global information across all variables.

Finally, we concatenate this global information with the preliminary forecast for each variable $\overline{Y}$, from the previous stage. The result is processed by a linear layer that performs a weighted fusion. By doing this, every variable can incorporate the global context from all other variables, allowing the model to better learn their correlations and improve the final forecast $\hat{Y}$. following these formulas:

$$\hat{Y} = \text{Linear}(\text{Concat}(\overline{Y}, \hat{X})) \text{ where } \overline{Y} = \text{Linear}(\text{LPF}(X)) \tag{2}$$

Table 1: Statistics of datasets for our experiments.

| Datasets | Weather | Electricity | Traffic | Solar | ETTh1 | ETTh2 | ETTm1 | ETTm2 |
|---|---|---|---|---|---|---|---|---|
| Features | 21 | 321 | 862 | 137 | 7 | 7 | 7 | 7 |
| Timesteps | 52696 | 26304 | 17544 | 52560 | 17420 | 17420 | 69680 | 69680 |
| Frequency | 10 min | Hourly | Hourly | 10 min | Hourly | Hourly | 15 min | 15 min |

## 4 EXPERIMENTS

### 4.1 DATASETS

To evaluate the forecasting capability of DCTS on real-world time series data, we conducted experiments on eight datasets, namely ETTm1, ETTm2, ETTh1, ETTh2, Weather, Solar, Traffic, and Electricity(Lai et al., 2018). Among these datasets are widely used for various benchmark tests. Table 1 presents the statistical data of these datasets.

### 4.2 BASELINES

We selected eight time series forecasting models for comparison with our model, including the Transformer-based iTransformer(Liu et al., 2023), PatchTST(Nie et al., 2022) and Crossformer(Zhang & Yan, 2023) models, linear model SOFTS(Han et al., 2024), TSMixer(Ekambaram et al., 2023), TiDE(Das et al., 2023), the TimesNet(Wu et al., 2022) model based on periodic decomposition, with SOFTS maintaining state-of-the-art performance in long-term time series forecasting.

### 4.3 EXPERIMENTAL SETUP

The experiments were conducted using an NVIDIA Tesla A40 48G GPU. We trained the models using Mean Squared Error (MSE) loss. The lookback window size for all models was set to $L = 96$,

Table 2: Forecast results with 96 review window and prediction length {96, 192, 336, 720}. The best result is represented in **red** bold, followed by bule underline. Results are averaged from all prediction lengths

| Models | | Ours | | SOFTS | | iTransformer | | PatchTST | | TSMixer | | Crossformer | | TiDE | | TimesNet | | DLinear | |
|---|---|---|---|---|---|---|---|---|---|---|---|---|---|---|---|---|---|---|---|
| Metric | | MSE | MAE | MSE | MAE | MSE | MAE | MSE | MAE | MSE | MAE | MSE | MAE | MSE | MAE | MSE | MAE | MSE | MAE |
| ETTm1 | 96 | 0.315 | 0.353 | 0.325 | 0.361 | 0.334 | 0.368 | 0.329 | 0.365 | 0.323 | 0.363 | 0.404 | 0.426 | 0.364 | 0.387 | 0.338 | 0.375 | 0.345 | 0.372 |
| | 192 | 0.360 | 0.378 | 0.375 | 0.389 | 0.377 | 0.391 | 0.380 | 0.394 | 0.376 | 0.392 | 0.450 | 0.451 | 0.398 | 0.404 | 0.374 | 0.387 | 0.380 | 0.389 |
| | 336 | 0.397 | 0.405 | 0.405 | 0.412 | 0.426 | 0.420 | 0.400 | 0.410 | 0.407 | 0.413 | 0.532 | 0.515 | 0.428 | 0.425 | 0.410 | 0.411 | 0.413 | 0.413 |
| | 720 | 0.470 | 0.447 | 0.466 | 0.447 | 0.491 | 0.459 | 0.475 | 0.453 | 0.485 | 0.459 | 0.666 | 0.589 | 0.487 | 0.461 | 0.478 | 0.450 | 0.474 | 0.453 |
| Avg | | 0.385 | 0.395 | 0.393 | 0.403 | 0.407 | 0.410 | 0.396 | 0.406 | 0.398 | 0.407 | 0.513 | 0.496 | 0.419 | 0.419 | 0.400 | 0.406 | 0.403 | 0.407 |
| ETTm2 | 96 | 0.172 | 0.256 | 0.180 | 0.261 | 0.180 | 0.264 | 0.184 | 0.264 | 0.182 | 0.266 | 0.287 | 0.366 | 0.207 | 0.305 | 0.187 | 0.267 | 0.193 | 0.292 |
| | 192 | 0.236 | 0.298 | 0.246 | 0.306 | 0.250 | 0.309 | 0.246 | 0.306 | 0.249 | 0.309 | 0.414 | 0.492 | 0.290 | 0.364 | 0.249 | 0.309 | 0.284 | 0.362 |
| | 336 | 0.300 | 0.338 | 0.319 | 0.352 | 0.311 | 0.348 | 0.308 | 0.346 | 0.309 | 0.347 | 0.597 | 0.542 | 0.377 | 0.422 | 0.321 | 0.351 | 0.369 | 0.427 |
| | 720 | 0.406 | 0.400 | 0.405 | 0.401 | 0.412 | 0.407 | 0.409 | 0.402 | 0.416 | 0.408 | 1.730 | 1.042 | 0.558 | 0.524 | 0.408 | 0.403 | 0.554 | 0.522 |
| Avg | | 0.278 | 0.323 | 0.287 | 0.330 | 0.288 | 0.332 | 0.287 | 0.330 | 0.289 | 0.333 | 0.757 | 0.610 | 0.358 | 0.404 | 0.291 | 0.333 | 0.350 | 0.401 |
| ETTh1 | 96 | 0.377 | 0.400 | 0.381 | 0.399 | 0.386 | 0.405 | 0.394 | 0.406 | 0.401 | 0.412 | 0.423 | 0.448 | 0.479 | 0.464 | 0.384 | 0.402 | 0.386 | 0.400 |
| | 192 | 0.431 | 0.428 | 0.435 | 0.431 | 0.441 | 0.436 | 0.440 | 0.435 | 0.452 | 0.442 | 0.471 | 0.474 | 0.525 | 0.492 | 0.436 | 0.429 | 0.437 | 0.432 |
| | 336 | 0.477 | 0.451 | 0.480 | 0.452 | 0.487 | 0.458 | 0.491 | 0.462 | 0.492 | 0.463 | 0.570 | 0.546 | 0.565 | 0.515 | 0.491 | 0.469 | 0.481 | 0.459 |
| | 720 | 0.496 | 0.480 | 0.499 | 0.488 | 0.503 | 0.491 | 0.487 | 0.479 | 0.507 | 0.490 | 0.653 | 0.621 | 0.594 | 0.558 | 0.521 | 0.500 | 0.519 | 0.516 |
| Avg | | 0.445 | 0.439 | 0.449 | 0.442 | 0.454 | 0.447 | 0.453 | 0.446 | 0.463 | 0.452 | 0.529 | 0.522 | 0.541 | 0.507 | 0.458 | 0.450 | 0.456 | 0.452 |
| ETTh2 | 96 | 0.295 | 0.347 | 0.297 | 0.347 | 0.297 | 0.349 | 0.288 | 0.340 | 0.319 | 0.361 | 0.745 | 0.584 | 0.400 | 0.440 | 0.340 | 0.374 | 0.333 | 0.387 |
| | 192 | 0.376 | 0.396 | 0.373 | 0.394 | 0.380 | 0.400 | 0.376 | 0.395 | 0.402 | 0.410 | 0.877 | 0.656 | 0.528 | 0.509 | 0.402 | 0.414 | 0.477 | 0.476 |
| | 336 | 0.418 | 0.430 | 0.410 | 0.426 | 0.428 | 0.432 | 0.440 | 0.451 | 0.444 | 0.446 | 1.043 | 0.731 | 0.643 | 0.571 | 0.452 | 0.452 | 0.594 | 0.541 |
| | 720 | 0.429 | 0.447 | 0.411 | 0.433 | 0.436 | 0.453 | 0.427 | 0.445 | 0.436 | 0.450 | 1.104 | 0.763 | 0.874 | 0.679 | 0.462 | 0.468 | 0.831 | 0.657 |
| Avg | | 0.379 | 0.405 | 0.373 | 0.400 | 0.383 | 0.407 | 0.385 | 0.410 | 0.401 | 0.417 | 0.942 | 0.684 | 0.611 | 0.550 | 0.414 | 0.427 | 0.559 | 0.515 |
| ECL | 96 | 0.136 | 0.231 | 0.143 | 0.233 | 0.148 | 0.240 | 0.164 | 0.251 | 0.157 | 0.260 | 0.219 | 0.314 | 0.237 | 0.329 | 0.168 | 0.272 | 0.197 | 0.282 |
| | 192 | 0.155 | 0.249 | 0.158 | 0.248 | 0.162 | 0.253 | 0.173 | 0.262 | 0.173 | 0.274 | 0.231 | 0.322 | 0.236 | 0.330 | 0.184 | 0.289 | 0.196 | 0.285 |
| | 336 | 0.166 | 0.262 | 0.178 | 0.269 | 0.178 | 0.269 | 0.190 | 0.279 | 0.192 | 0.295 | 0.246 | 0.337 | 0.249 | 0.344 | 0.198 | 0.300 | 0.209 | 0.301 |
| | 720 | 0.191 | 0.287 | 0.218 | 0.305 | 0.225 | 0.317 | 0.230 | 0.313 | 0.223 | 0.318 | 0.280 | 0.363 | 0.284 | 0.373 | 0.220 | 0.320 | 0.245 | 0.333 |
| Avg | | 0.162 | 0.257 | 0.174 | 0.264 | 0.178 | 0.270 | 0.189 | 0.276 | 0.186 | 0.287 | 0.244 | 0.334 | 0.251 | 0.344 | 0.192 | 0.295 | 0.212 | 0.300 |
| Traffic | 96 | 0.407 | 0.256 | 0.376 | 0.251 | 0.395 | 0.268 | 0.427 | 0.272 | 0.493 | 0.336 | 0.522 | 0.290 | 0.805 | 0.493 | 0.593 | 0.321 | 0.650 | 0.396 |
| | 192 | 0.443 | 0.270 | 0.398 | 0.261 | 0.417 | 0.276 | 0.454 | 0.289 | 0.497 | 0.351 | 0.530 | 0.293 | 0.756 | 0.474 | 0.617 | 0.336 | 0.598 | 0.370 |
| | 336 | 0.466 | 0.287 | 0.415 | 0.269 | 0.433 | 0.283 | 0.450 | 0.282 | 0.528 | 0.361 | 0.558 | 0.305 | 0.762 | 0.477 | 0.629 | 0.336 | 0.605 | 0.373 |
| | 720 | 0.504 | 0.298 | 0.447 | 0.287 | 0.467 | 0.302 | 0.484 | 0.301 | 0.569 | 0.380 | 0.589 | 0.328 | 0.719 | 0.449 | 0.640 | 0.350 | 0.645 | 0.394 |
| Avg | | 0.455 | 0.277 | 0.409 | 0.267 | 0.428 | 0.282 | 0.454 | 0.285 | 0.522 | 0.357 | 0.550 | 0.304 | 0.760 | 0.473 | 0.620 | 0.336 | 0.625 | 0.383 |
| Weather | 96 | 0.155 | 0.200 | 0.166 | 0.208 | 0.174 | 0.214 | 0.176 | 0.217 | 0.166 | 0.210 | 0.158 | 0.230 | 0.202 | 0.261 | 0.172 | 0.220 | 0.196 | 0.255 |
| | 192 | 0.206 | 0.248 | 0.217 | 0.253 | 0.221 | 0.254 | 0.221 | 0.256 | 0.215 | 0.256 | 0.206 | 0.277 | 0.242 | 0.298 | 0.219 | 0.261 | 0.237 | 0.296 |
| | 336 | 0.262 | 0.289 | 0.282 | 0.300 | 0.278 | 0.296 | 0.275 | 0.296 | 0.287 | 0.300 | 0.272 | 0.335 | 0.287 | 0.335 | 0.280 | 0.306 | 0.283 | 0.335 |
| | 720 | 0.348 | 0.344 | 0.356 | 0.351 | 0.358 | 0.347 | 0.352 | 0.346 | 0.355 | 0.348 | 0.398 | 0.418 | 0.351 | 0.386 | 0.365 | 0.359 | 0.345 | 0.381 |
| Avg | | 0.242 | 0.270 | 0.255 | 0.278 | 0.258 | 0.278 | 0.256 | 0.279 | 0.256 | 0.279 | 0.259 | 0.315 | 0.271 | 0.320 | 0.259 | 0.287 | 0.265 | 0.317 |
| Solar | 96 | 0.197 | 0.234 | 0.200 | 0.230 | 0.203 | 0.237 | 0.205 | 0.246 | 0.221 | 0.275 | 0.310 | 0.331 | 0.312 | 0.399 | 0.250 | 0.292 | 0.290 | 0.378 |
| | 192 | 0.226 | 0.258 | 0.229 | 0.253 | 0.233 | 0.261 | 0.237 | 0.267 | 0.268 | 0.306 | 0.734 | 0.725 | 0.339 | 0.416 | 0.296 | 0.318 | 0.320 | 0.398 |
| | 336 | 0.237 | 0.269 | 0.243 | 0.269 | 0.248 | 0.273 | 0.250 | 0.276 | 0.272 | 0.294 | 0.750 | 0.735 | 0.368 | 0.430 | 0.319 | 0.330 | 0.353 | 0.415 |
| | 720 | 0.245 | 0.275 | 0.245 | 0.272 | 0.249 | 0.275 | 0.252 | 0.275 | 0.281 | 0.313 | 0.769 | 0.765 | 0.370 | 0.425 | 0.338 | 0.337 | 0.356 | 0.413 |
| Avg | | 0.226 | 0.259 | 0.229 | 0.256 | 0.233 | 0.262 | 0.236 | 0.266 | 0.260 | 0.297 | 0.641 | 0.639 | 0.347 | 0.417 | 0.301 | 0.319 | 0.330 | 0.401 |

with prediction lengths of $T = \{96, 192, 336, 720\}$. For the baselines, data related to SOFTS was utilized. The specific experimental data are presented in Table 2.

## 4.4 RESULTS AND ANALYSIS

Table 2 summarizes the forecasting performance of all methods on eight real-world time series datasets, demonstrating the superior performance of DCTS. Specifically, across different prediction lengths, it achieved the best performance on six datasets and the second-best performance on two dataset. On the six datasets where it surpasses the SOTA model SOFTS, DCTS achieves performance improvements of 2.7% (in MSE) and 1.5% (in MAE). Although SOFTS also samples the time series representation, its performance is not as good because it only focuses on continuous encoding, and because of the strong representation ability of continuous encoding, the model is prone to learn more noise. This makes the sampled information lack the discrete patterns in the time series. Meanwhile, the channel-independent mechanism used by PatchTST, while improving robustness, also loses the correlation information between variables.

## 4.5 ABLATION ANALYSIS

This section presents several ablation studies to validate the design of DCTS. We conduct two main ablation experiments and perform evaluations on four datasets (ETTh1, ETTh2, ETTm1, and ETTm2). The following are the explanations for each variant:

1. **w/o-D:** In this variant, we remove the Discrete Module. The embedding process relies solely on the original continuous information for encoding.

2. **w/o-C:** In this variant, we remove the Continuous Module. The model only encodes the time series discretely, and the decoder's output is directly projected to the forecast horizon to be the final prediction.

Table 3 shows the results of all ablation studies. Specifically, our summary and explanation of the two ablation experiments are as follows:

1. **Discrete Module:** For this experiment, we removed the Discrete Module, relying solely on continuous information for time series encoding. The results indicate that incorporating discrete encoding allows the model to capture diverse patterns in the time series, leading to an average performance gain of 2.1% in MSE and 1.3% in MAE.

2. **Continuous Module:** For this experiment, we removed the Continuous Module and relied solely on the Discrete Module for forecasting. As the results show, this caused a drastic decline in performance: a 16.8% drop in MSE and an 11.7% drop in MAE. The reason is that time series data consists mainly of continuous information because it represents the changes in a dynamic system. Relying only on discrete patterns to represent the time series causes a significant loss of detail, which in turn degrades the model's performance.

Table 3: The table above shows the ablation results for ETT. The results represent the loss of MSE and MAE for forecast lengths {96, 192, 336, 720}, with the best results highlighted in bold.

| Model | | T=96 | | T=192 | | T=336 | | T=720 | | Avg | |
|---|---|---|---|---|---|---|---|---|---|---|---|
| Metric | | MSE | MAE | MSE | MAE | MSE | MAE | MSE | MAE | MSE | MAE |
| **Ours** | ETTh1 | **0.377** | **0.400** | **0.431** | **0.428** | **0.477** | **0.451** | **0.496** | **0.480** | **0.445** | **0.439** |
| | ETTh2 | **0.295** | **0.347** | **0.376** | **0.396** | **0.418** | **0.430** | **0.429** | **0.447** | **0.379** | **0.405** |
| | ETTm1 | **0.315** | **0.353** | **0.360** | **0.378** | **0.397** | **0.405** | **0.470** | **0.447** | **0.385** | **0.395** |
| | ETTm2 | **0.172** | **0.256** | **0.236** | **0.298** | **0.300** | **0.338** | **0.406** | **0.400** | **0.278** | **0.323** |
| **w/o-D** | ETTh1 | 0.384 | 0.402 | 0.440 | 0.432 | 0.485 | 0.458 | 0.499 | 0.487 | 0.452 | 0.444 |
| | ETTh2 | 0.297 | 0.348 | 0.376 | 0.396 | 0.419 | 0.431 | 0.449 | 0.458 | 0.385 | 0.408 |
| | ETTm1 | 0.330 | 0.366 | 0.371 | 0.386 | 0.407 | 0.410 | 0.475 | 0.449 | 0.395 | 0.402 |
| | ETTm2 | 0.177 | 0.260 | 0.245 | 0.306 | 0.309 | 0.347 | 0.413 | 0.405 | 0.286 | 0.329 |
| **w/o-C** | ETTh1 | 0.461 | 0.458 | 0.563 | 0.511 | 0.594 | 0.530 | 0.551 | 0.519 | 0.542 | 0.504 |
| | ETTh2 | 0.352 | 0.386 | 0.428 | 0.429 | 0.455 | 0.455 | 0.466 | 0.468 | 0.425 | 0.434 |
| | ETTm1 | 0.459 | 0.451 | 0.484 | 0.461 | 0.510 | 0.476 | 0.564 | 0.502 | 0.504 | 0.472 |
| | ETTm2 | 0.211 | 0.293 | 0.284 | 0.338 | 0.338 | 0.396 | 0.433 | 0.419 | 0.316 | 0.361 |

## 4.6 CONTRASTIVE ANALYSIS

Traditional vector quantization methods use a single codebook to discretize data. Each data point is assigned a single, unique discrete code. This method of using a single discrete code struggles to cover the entire feature space of a time series. Therefore, we use a multi-codebook approach to exponentially expand the discrete space.

In Figure 3, we compare the performance of single-codebook and multi-codebook models on the ETTm1, ETTm2, Electricity and Solar datasets. The figure shows that for the same individual codebook size, introducing multiple codebooks improves model performance compared to using a single one. Across all prediction windows, the average performance improved on the Solar dataset by 9.6% in MSE and 6.2% in MAE. On the Electricity dataset, it improved by 1.2% in MSE and 0.8% in MAE. On ETTm2, the improvement was 1.1% in MSE and 0.9% in MAE, and on ETTm1, it was 0.5% in MSE and 0.5% in MAE. Additionally, the multi-codebook method changes the data's representation from a single "letter" to a "word." This allows us to use smaller codebooks that work together to create a joint representation, achieving a larger encoding space than a single codebook.

To prove that adding discrete information to continuous models is necessary, we added our Discrete Module to other continuous models. We chose PatchTST and iTransformer as our baseline models. The detailed experimental results are in Table4. We conducted these comparative experiments on the ETTm1 and ETTm2 datasets. The models "PatchTST+VQ" and "iTrans+VQ" represent the

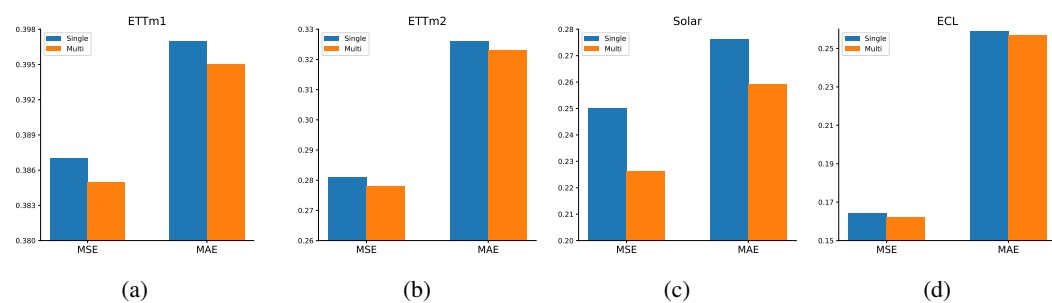

(a)        (b)        (c)        (d)

Figure 3: On the ETTm1, ETTm2, Electricity and Solar dataset, we presented the comparison results of single-codebook and multi-codebook approaches. The result is the average value of the predicted length of {96, 192, 336, 720}.

original PatchTST and iTransformer with discrete information module added. We did not change the parameters of the original backbone models.

The findings indicate that adding discrete encoding boosts performance for all forecast horizons on both datasets. Across all experiments, the average improvement for PatchTST was 3.5% (MSE) and 2.7% (MAE), while for iTransformer it was 2.3% (MSE) and 1.3% (MAE). This demonstrates that even though time series data is mainly continuous, jointly modeling both discrete and continuous information enhances a model's expressive capabilities. Therefore, discrete coding is necessary for time series analysis.

Table 4: The table above shows the contrast results for add discrete information with original. The results represent the loss of MSE and MAE for forecast lengths {96, 192, 336, 720}, with the best results highlighted in bold.

| Model | | **PatchTST+VQ** | | PatchTST | | **iTrans+VQ** | | iTrans | |
|---|---|---|---|---|---|---|---|---|---|
| Metric | | MSE | MAE | MSE | MAE | MSE | MAE | MSE | MAE |
| ETTm1 | 96 | **0.325** | **0.362** | 0.329 | 0.365 | **0.326** | **0.364** | 0.334 | 0.368 |
| | 192 | **0.361** | **0.380** | 0.380 | 0.394 | **0.370** | **0.388** | 0.377 | 0.391 |
| | 336 | **0.390** | **0.407** | 0.400 | 0.410 | **0.409** | **0.413** | 0.426 | 0.420 |
| | 720 | **0.459** | **0.443** | 0.475 | 0.453 | **0.476** | **0.450** | 0.491 | 0.459 |
| Avg | | **0.383** | **0.398** | 0.396 | 0.406 | **0.395** | **0.404** | 0.407 | 0.410 |
| ETTm2 | 96 | **0.171** | **0.253** | 0.184 | 0.264 | **0.175** | **0.258** | 0.180 | 0.264 |
| | 192 | **0.237** | **0.296** | 0.246 | 0.306 | **0.247** | **0.308** | 0.250 | 0.309 |
| | 336 | **0.300** | **0.337** | 0.308 | 0.346 | **0.308** | **0.346** | 0.311 | 0.348 |
| | 720 | **0.392** | **0.392** | 0.409 | 0.402 | **0.408** | **0.404** | 0.412 | 0.407 |
| Avg | | **0.275** | **0.319** | 0.287 | 0.330 | **0.284** | **0.329** | 0.288 | 0.332 |

### 4.7 CASES OF VISUALIZATION

In this section, we present visual examples of the discrete encodings to demonstrate our approach. As illustrated in Figure 4, we show how one codebook assigns discrete codes to six different time series samples from the ETTh1 dataset. Samples marked with the same color have been assigned the same discrete code.

It is clear that different discrete codes correspond to different time series patterns. As we can observe, the samples in Figure 4b mostly include four troughs, while those in Figure 4b primarily feature four peaks. The samples in Figure 4c, however, are comparatively more stable and show fewer fluctuations than the others. This demonstrates that the different vectors within one codebook

learn distinct time series patterns. This allows the codebook to assign a specific discrete code to any variable that matches one of the learned patterns.

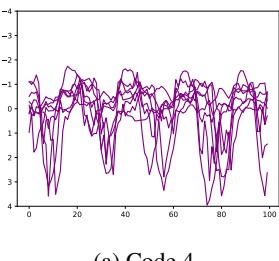 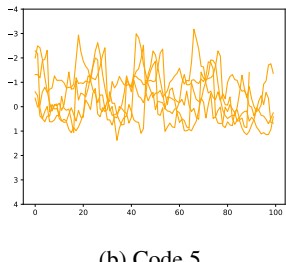 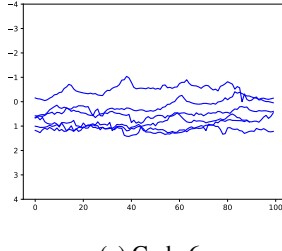

(a) Code 4            (b) Code 5            (c) Code 6

Figure 4: One of the codebooks performs discrete encoding on different samples of ETTh1. For the same color, the discrete encoding is the same.

On the ETTm2 dataset, we also visualized the impact of discrete encoding on model performance. As illustrated in Figure 5, we incorporated discrete encoding into the PatchTST model and compared it against the original version. We can see that compared to the original model, the predictions of the model with discrete encoding are closer to the ground truth. Furthermore, the patterns in the forecast align better with the ground truth patterns. A model with only continuous encoding can be so expressive that it over-focuses on minor details, which weakens its ability to identify the main underlying patterns. By adding discrete encoding, the model's pattern-learning capabilities are strengthened. This combination ensures that the final predictions are both expressive and better at recognizing overall patterns.

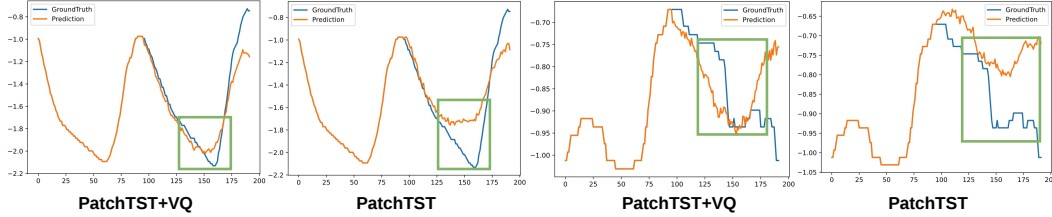

Figure 5: The above figure shows the comparison between the output visualization results of the PatchTST model with discrete encoding and the original model.

## 5 CONCLUSION

In this work, we introduced DCTS to address the issues that arise from using purely continuous encoding in time series forecasting. Our model combines continuous and discrete encoding, giving it the high expressive power of continuous representations as well as the ability of discrete codes to learn overall patterns. Extensive experiments on real-world datasets demonstrate that DCTS is superior to existing models for forecasting tasks. Furthermore, our ablation and comparative studies confirm that discrete encoding is a necessary component for time series analysis.

Looking ahead, one limitation of current vector quantization is its reliance on Euclidean distance for similarity. When a time series sample is phase-shifted, this distance metric can be easily increased, which may lead to the sample being assigned the wrong discrete code. Dynamic Time Warping (DTW) is an algorithm that also measures the similarity between sequences. Unlike Euclidean distance and cosine similarity, DTW is more focused on the shape similarity of time series, which can make it more effective for handling samples that are shifted in time. The trade-off, however, is that DTW is not very efficient. Its computation time is significantly longer than that of both Euclidean distance and cosine similarity. A key area for future work, therefore, is to develop better methods for quantizing time series data.

ETHICS STATEMENT

This research complies with the ICLR ethical guidelines. Our work does not involve any experiments on humans or animals. All datasets used in this study are public, and we have followed their usage guidelines to protect privacy. We have avoided any biased or discriminatory results. No personally identifiable information was used, nor did we conduct any experiments that could pose privacy or security risks. We have maintained transparency and integrity throughout this research.

REPRODUCIBILITY STATEMENT

We ensure the reproducibility of the results in this paper. Part of our code and all of our running scripts are publicly available in the supplementary materials. We have provided detailed descriptions of our experimental setup and hardware specifications. All datasets used are public, which ensures that our results are consistent and repeatable. We believe these measures will enable other researchers to replicate our work and help advance the field.

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

## A APPENDIX

### A.1 LLM USAGES

Large Language Models (LLMs) were used to assist in writing and refining this manuscript. Specifically, we used LLMs to improve the language, enhance the paper's readability, and ensure the clarity of each section. The image of the cat in Figure 1 was also generated by a generative model, but this was for illustrative purposes only and is not related to our research methodology. The LLMs were not involved in the conceptualization, research methods, or experimental design of this paper. All concepts, ideas, and analyses were developed and implemented solely by the authors. The contribution of the LLMs was strictly limited to improving the linguistic quality of the text; they played no role in the scientific content or data analysis. The authors take full responsibility for the content of this manuscript. We have ensured that any text generated by the LLM adheres to ethical guidelines and is free from plagiarism and academic misconduct.

### A.2 HYPERPARAMETER SENSITIVITY EXPERIMENTS

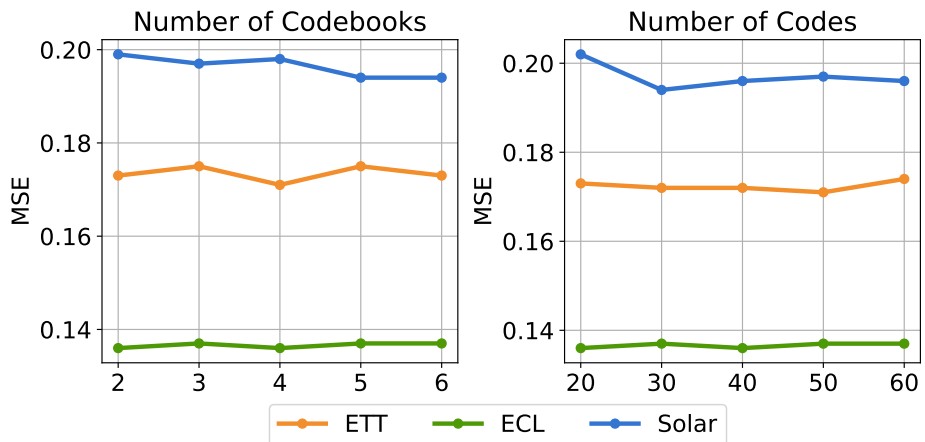

Figure 6: Hyperparameter Sensitivity Experiments

We provided a sensitivity analysis for the hyperparameters. We conducted experiments on multiple datasets for the most important parameters: the number of codebooks and the codebook size in Figure 6. The experiments show that these hyperparameters have a minimal impact on the model,

demonstrating its robustness and generalization. For the Solar dataset, which is more complex than others due to factors like weather (e.g., cloud cover), increasing the number and size of the codebooks can improve performance. However, the improvement is not significant. Conversely, using more and larger codebooks will reduce the model's computational efficiency.

## A.3 INPUT LENGTHS EXPERIMENTS

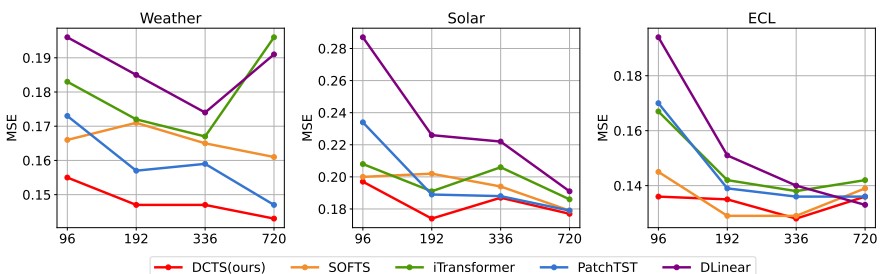

Figure 7: Experiments of different input lengths on prediction results

We provided the performance of different models across various history lengths in Figure 7. The results show that our model achieves the best performance across most datasets and input lengths. This demonstrates that our model performs well with different input lengths.

## A.4 CODE VISUALIZATION

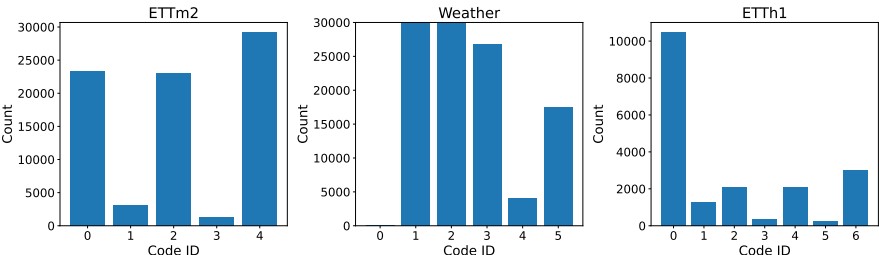

Figure 8: Counts of code used

We provided the codebook usage across different datasets in Figure 8. In a reasonably sized codebook, although some codes are used less frequently, there are no dead codes. However, when the codebook size is too small, the codes have great distinctiveness. In a steady dataset, all samples might aggregate into a single code. When the codebook size is too large, some codes might not be used at all. Both situations will lead to the creation of dead codes.