# OpenReview forum: "DCTS: Fusing Discrete and Continuous Information for Time Series Forecasting"
_ICLR.cc/2026/Conference — Submitted to ICLR 2026_

### Official Review · Reviewer_bN1W · 2025-10-24

**Soundness:** 3
**Presentation:** 2
**Contribution:** 3
**Rating:** 4
**Confidence:** 3

**Summary:**

This paper proposes a time series forecasting method that combines discrete encodings with continuous encodings. The contributions of this paper are as follows:
1. Proposes fusing discrete and continuous information for time series forecasting (DCTS)
2. Attempts to replace single codebook with multiple codebooks to encode time series.
The proposed method outperforms baseline methods on multiple real-world datasets.

**Strengths:**

Originality: This is the first work trying to combine the advantages of both continuous and discrete encodings in time series forecasting. Though the idea is kind of straight, this paper still deserves originality.
Quality: Overall the solution is clearly motivated and reasonably implemented, and corresponding evaluations are comprehensive.
Clarity: The paper is easy to read, the results are easy to understand.
Significance: This study integrates continuous and discrete encoding characteristics, achieving improved performance compared to current purely continuous-encoding time series prediction models.

**Weaknesses:**

1. The performance comparison could be strengthened by including a more contemporary set of baseline models. The selection appears somewhat conservative, with only one model (SOFTS) published after 2024. While the ablation studies effectively demonstrate the benefit of combining continuous and discrete encodings over either alone, the primary goal remains advancing time series forecasting performance. Therefore, comparisons with other recent open-source works (e.g., CycleNet) would be more compelling. Furthermore, to more directly validate the superiority of the proposed hybrid approach, a comparison against a strong discrete-based baseline—such as by adapting SDFormer to the time series forecasting task—would be highly informative.
2. The manuscript contained a number of grammatical issues and non-idiomatic expressions that hindered understanding. I strongly recommend thorough language polishing to ensure the writing's clarity matches its technical quality.

If my above concerns are resolved, I would consider increasing my rating.

**Questions:**

My questions are listed above.

---

> ### Author Response · Authors · 2025-11-21
>
> Thank you very much for your detailed review and valuable feedback on our paper.
>
> We compared our results with TimeXer, Leddam, and CycleNet. As shown in the table below, our method still achieves the best performance, even when compared to these state-of-the-art models.
>
> | Models || **Ours** || TimeXer || Leddam || CycleNet ||
> | :--- | :---: | :---: | :---: | :---: | :---: | :---: | :---: | :---: | :---: |
> | Metric || MSE | MAE | MSE | MAE | MSE | MAE | MSE | MAE |
> | **ECL** | 96 | **0.136** | **0.231** | 0.140 | 0.242 | 0.141 | 0.235 | 0.141 | 0.234 |
> | | 192 | **0.155** | **0.249** | 0.157 | 0.256 | 0.159 | 0.252 | 0.155 | 0.247 |
> | | 336 | **0.166** | **0.262** | 0.176 | 0.275 | 0.173 | 0.268 | 0.172 | 0.264 |
> | | 720 | **0.191** | **0.287** | 0.211 | 0.306 | 0.201 | 0.295 | 0.210 | 0.296 |
> | **Traffic** | 96 | **0.407** | **0.256** | 0.428 | 0.271 | 0.426 | 0.276 | 0.458 | 0.296 |
> | | 192 | **0.443** | **0.270** | 0.448 | 0.282 | 0.458 | 0.289 | 0.457 | 0.294 |
> | | 336 | **0.466** | **0.287** | 0.473 | 0.289 | 0.486 | 0.297 | 0.470 | 0.299 |
> | | 720 | 0.504 | **0.298** | 0.516 | 0.307 | **0.498** | 0.313 | 0.502 | 0.314 |
>
> Since SDFormer does not adopt a unified framework for time series prediction and does not provide the ETT dataset it used and the specific experimental settings, it will take a little more time to modify it for application in the time series prediction task. We have provided a comparative experiment with another method in the field of time series prediction that uses VQ, Sparse-VQ. Compared with Sparse-VQ, we can achieve better results.
>
> | Models ||**Ours**| | Sparse-VQ||
> | :--- | :---: | :---: | :---: | :---: | :---: |
> | Metric || MSE | MAE | MSE | MAE |
> | **ETTh1** | 96 | **0.056** | **0.182** | 0.056 | 0.184 |
> | | 192 | **0.071** | **0.203** | 0.072 | 0.210 |
> | | 336 | **0.082** | **0.221** | 0.079 | 0.224 |
> | | 720 | **0.080** | **0.226** | 0.084 | 0.231 |
> | **ETTh2** | 96 | **0.133** | **0.281** | 0.133 | 0.283 |
> | | 192 | 0.178 | 0.338 | **0.174** | **0.331** |
> | | 336 | **0.181** | **0.341** | 0.181 | 0.343 |
> | | 720 | **0.183** | **0.348** | 0.215 | 0.372 |

---

> ### Author Response · Authors · 2025-12-03
>
> We provided an experiment where SDformer was applied to the forecasting task. Although SDformer is a powerful generative model, it is not suitable for long-term multivariate time series forecasting.
> | Models |  | Ours |  | SDformer  |  |
> | :--- | :--- | :--- | :--- | :--- | :--- |
> |Metric||MSE|MAE|MSE|MAE|
> | ETTm1 | 96 | 0.315 | 0.353 | 1.105 | 0.792 |
> | | 192 | 0.360 | 0.378 | 1.104 | 0.791 |
> | | 336 | 0.397 | 0.405 | 1.105 | 0.793 |
> | | 720 | 0.470 | 0.447 | 1.107 | 0.796 |
> | ETTh1 | 96 | 0.377 | 0.400 | 1.111 | 0.798 |
> | | 192 | 0.431 | 0.428 | 1.115 | 0.800 |
> | | 336 | 0.477 | 0.451 | 1.110 | 0.802 |
> | | 720 | 0.496 | 0.480 | 1.104 | 0.808 |

---

### Official Review · Reviewer_mLa9 · 2025-10-29

**Soundness:** 2
**Presentation:** 2
**Contribution:** 2
**Rating:** 2
**Confidence:** 5

**Summary:**

This paper introduces an innovative time series forecasting model, DCTS (Discrete and Continuous Time Series Fusion), which combines discrete encoding and continuous encoding to enhance both pattern abstraction and numerical expression abilities.
Traditional time series models typically focus solely on continuous features, ignoring the underlying discrete patterns in the data (such as workday/weekend or peak/off-peak characteristics). This can lead to overfitting to noise or neglecting key patterns. DCTS addresses this issue by introducing vector quantization (VQ) into continuous models, leveraging a multi-codebook structure to generate multi-dimensional discrete codes, significantly enhancing the feature space’s expressiveness. The model consists of Discrete Module and Continuous Module. Experiments on multiple real-world datasets, including ETT, Weather, Traffic, Electricity, and Solar, show that DCTS outperforms state-of-the-art models (e.g., SOFTS, PatchTST, iTransformer) in terms of both MSE and MAE. Ablation studies confirm the complementarity of the two modules, while contrastive experiments highlight the advantages of the multi-codebook mechanism for capturing complex time series patterns.

**Strengths:**

1. DCTS is the first model to integrate discrete vector quantization with continuous time series modeling, utilizing multi-codebooks to expand the discrete encoding space.
2.  DCTS improved MSE by 2.7% and MAE by 1.5% compared to state-of-the-art models, demonstrating its effectiveness and generalization ability
3. The proposed dual-module structure (Discrete + Continuous) clearly separates the responsibilities: the discrete module captures global patterns, while the continuous module focuses on fine-grained changes.

**Weaknesses:**

1. Except for SOFT, all compared baselines are from 2023 or earlier, lacking comparisons with the most recent works.
2. The performance improvement achieved by the model is minimal and remains lower than that of some more advanced baselines[1,2].
3. There is no analysis of key hyperparameters. In addition, the input length is fixed at 96, but many of the compared baselines (such as PatchTST and DLinear) are generally not suitable for this setting. Overall, the experimental section is incomplete and insufficient.
4. The core technology mainly consists of a combination of existing methods. Moreover, Vector Quantization has already been explored in prior studies [3,4]. Therefore, the technical innovation of this paper is limited.
5. Since the authors emphasize the advantages of using a codebook, the following comparative experiments should be conducted: replace the codebook used in the proposed model with traditional patch or adaptive patch techniques [5], or alternatively, use simple temporal and variable encoding methods [6].
6. There is a lack of visualization for the codebook, making it impossible for reviewers to determine whether issues such as dead codes exist.
7. Since the authors emphasize that the proposed method can help the model recognize certain pattern characteristics of data across different time periods, and the example used is traffic data, corresponding experiments should be conducted to validate this motivation. As shown in paper [7], can the method accurately predict sudden events in datasets such as PEMS or METR-LA?
8. In summary, I believe that the experimental results presented in the current version are insufficient. The proposed method is neither solid nor innovative. I recommend that the authors conduct a comprehensive revision, particularly by adding experiments that validate the motivation and by including comparisons with more advanced baselines.

[1]  TimeXer: Empowering Transformers for Time Series Forecasting with Exogenous Variables

[2] Revitalizing Multivariate Time Series Forecasting: Learnable Decomposition with Inter-Series Dependencies and Intra-Series Variations Modeling

[3] Sparse-vq transformer: An ffn-free framework with vector quantization for enhanced time series forecasting

[4] Fusionsf: Fuse heterogeneous modalities in a vector quantized framework for robust solar power forecasting

[5] Enhancing Time Series Forecasting through Selective Representation Spaces: A Patch Perspective

[6] Spatial-temporal identity: A simple yet effective baseline for multivariate time series forecasting

[7] Hutformer: Hierarchical u-net transformer for long-term traffic forecasting

**Questions:**

See weakness

---

> ### Author Response · Authors · 2025-11-21
>
> Thank you very much for your review and valuable feedback on our paper.
>
> W1, 2: We provided a comparison with TimeXer and Leddam. Our model still achieves the best performance.
>
> | Models ||**Ours**|| TimeXer || Leddam ||
> | :--- | :---: | :---: | :---: | :---: | :---: | :---: | :---: |
> | Metric || MSE | MAE | MSE | MAE | MSE | MAE |
> | **ECL** | 96 | **0.136** | **0.231** | 0.140 | 0.242 | 0.141 | 0.235 |
> | | 192 | **0.155** | **0.249** | 0.157 | 0.256 | 0.159 | 0.252 |
> | | 336 | **0.166** | **0.262** | 0.176 | 0.275 | 0.173 | 0.268 |
> | | 720 | **0.191** | **0.287** | 0.211 | 0.306 | 0.201 | 0.295 |
> | **Traffic** | 96 | **0.407** | **0.256** | 0.428 | 0.271 | 0.426 | 0.276 |
> | | 192 | **0.443** | **0.270** | 0.448 | 0.282 | 0.458 | 0.289 |
> | | 336 | **0.466** | **0.287** | 0.473 | 0.289 | 0.486 | 0.297 |
> | | 720 | 0.504 | **0.298** | 0.516 | 0.307 | **0.498** | 0.313 |
>
> W3: We provided a sensitivity analysis for the hyperparameters. We conducted experiments on multiple datasets for the most important parameters: the number of codebooks and the codebook size in A.2 Figure 6. Setting the historical sequence length to 96 for comparison is a fair approach, as different lengths contain significantly different amounts of information. This is the comparison method chosen by major open-source projects like SOFTS and iTransformer. We provided the performance of different models across various history lengths in A.3 Figure 7. The results show that our model achieves the best performance across most datasets and input lengths.
>
> W4: Our core innovation is not the first use of VQ in time series forecasting. Instead, we propose a method that mix discrete and continuous representation. This improves prediction performance by combining the ability of discrete representation to capture global patterns with the ability of continuous representation to capture detailed information. Sparse-VQ aims to eliminate data noise using sparse vectors, while Fusionsf uses VQ to combine multimodal data. Thus, our method is fundamentally different from both.
>
> W5: We provided the prediction results after removing the codebooks. Model performance drops significantly without the codebooks, which proves that the codebooks are essential for the model.
>
> | Models ||**Original** || w/o-codebook ||
> | :--- | :---: | :---: | :---: | :---: | :---: |
> | Metric || MSE | MAE | MSE | MAE |
> || ETTh1 | **0.445** | **0.439** | 0.450 | 0.441 |
> | |ETTh2 | **0.379** | **0.405** | 0.397 | 0.413 |
> | |ETTm1 | **0.385** | **0.395** | 0.394 | 0.402 |
> | |ETTm2 | **0.278** | **0.323** | 0.283 | 0.324 |
>
> W6: We provided the codebook usage across different datasets in A.4 Figure 8. In a reasonably sized codebook, although some codes are used less frequently, there are no dead codes. However, when the codebook size is too small, the codes have great distinctiveness. In a steady dataset, all samples might aggregate into a single code. When the codebook size is too large, some codes might not be used at all. Both situations will lead to the creation of dead codes.
>
> W7: We provided the prediction results for the PEMS dataset. Even with this complex dataset, our method still achieves good results.
>
> | Models || **Ours** || SOFTS || iTrans ||
> | :--- | :---: | :---: | :---: | :---: | :---: | :---: | :---: |
> | Metric || MSE | MAE | MSE | MAE | MSE | MAE |
> | **PEMS03** | 12 | **0.062** | **0.165** | 0.064 | 0.165 | 0.071 | 0.174 |
> | | 24 | **0.078** | **0.184** | 0.083 | 0.188 | 0.093 | 0.201 |
> | | 48 | **0.111** | **0.218** | 0.114 | 0.223 | 0.125 | 0.236 |
> | | 96 | 0.162 | 0.267 | **0.156** | **0.264** | 0.164 | 0.275 |
> | **PEMS04** | 12 | **0.071** | **0.175** | 0.074 | 0.176 | 0.078 | 0.183 |
> | | 24 | **0.080** | **0.184** | 0.088 | 0.194 | 0.095 | 0.205 |
> | | 48 | **0.100** | **0.208** | 0.110 | 0.219 | 0.120 | 0.233 |
> | | 96 | **0.133** | **0.240** | 0.135 | 0.244 | 0.150 | 0.262 |

---

### Official Review · Reviewer_LGZf · 2025-10-31

**Soundness:** 3
**Presentation:** 2
**Contribution:** 2
**Rating:** 4
**Confidence:** 4

**Summary:**

This paper advocates emphasizing discrete representations for time series, extracting discrete features via Vector Quantization (VQ), and introducing a multi-codebook (multi-vocabulary) mechanism. Through comprehensive experiments, it demonstrates the importance of fusing discrete and continuous representations for time series forecasting.

**Strengths:**

1. The manuscript proposes a method based on Vector Quantization (VQ) method that fuses discrete and continuous features, thereby improving its performance.
2. DCTS achieves state-of-the-art results by extensive experiments on many real-world datasets, demonstrating the effectiveness and validity.

**Weaknesses:**

1. One of the main innovations—using a multi-codebook for representation to enhance the representation ability. Existing work on representation learning has thoroughly explored this approach, such as RVQ [1] and Group-Residual VQ [2].
2. Regarding the claimed advantages of discrete encoding over continuous encoding in the introduction: in practice, periodic encodings (e.g., the sinusoidal positional encodings commonly used in Transformer models) already ensure that Monday follows Sunday, avoiding boundary issues. Moreover, many models assign an individual embedding to each time step for time encoding, which is also unordered and does not induce large artificial distances between adjacent days.
3. The experiment is somewhat weak. It lacks the results of the latest baselines of time series forecasting in 2025. Besides, in contrastive analysis, the manuscript omits hyperparameter sensitivity analyses, especially the impact of the number of codebooks. Meanwhile, the contrast results for add VQ can be more abundant, including the results of other baselines and datasets.
4. In the ablation study, removing the entire CI or CD module makes it unclear whether the encoder–decoder and frequency-domain components are driving the performance improvement. To convincingly show the value of the discrete and continuous parts, a core ablation should construct the fused representation X using only the discrete or only the continuous representation, while keeping all other network components within the CI and CD modules unchanged.

**Questions:**

See Weaknesses.

---

> ### Author Response · Authors · 2025-11-21
>
> Thank you for your detailed review and suggestions for our work.
>
> W1: Using multiple codebooks for the discrete representation of time series is a contribution of our work. Although RVQ and Group-Residual VQ also use multiple codebooks, their methods focus on  "Residuals". The codes learned through residuals are correlated. Each code is generated by re-encoding the residual of the input and the previous discrete code. While this approach can expand the code space, the codes cannot be learned independently. Our method enables each codebook to learn independently (similar to multi-head attention). Each discrete code can learn different discrete features, and together they form the discrete encoding for the time series.
>
> W2: Periodic encodings, like sinusoidal positional encoding, can address the boundary problem to some extent. However, as this is a fixed encoding method, it struggles to capture specific semantic meanings (such as the morning rush hour pattern on weekdays), even though it captures relative positions and periodicity. Assigning a separate embedding for each time step is essentially another discretization method. However, in fine-grained contexts (like by the hour or minute), this would create an extremely large embedding table. This makes it difficult for the model to learn time-specific relationships. Our vector quantization method balances these two issues through learning. It ensures the model can learn different time series patterns and achieves a larger code space using multiple codebooks without taking up more space.
>
> W3: We have added two new baseline models: TimeKAN (ICLR 2025) and DUET (KDD 2025). As shown in the table, our method still achieves the best performance, even when compared to these state-of-the-art models.
> | Models || **Ours** || DUET || TimeKAN ||
> | :--- | :---: | :---: | :---: | :---: | :---: | :---: | :---: |
> | Metric || MSE | MAE | MSE | MAE | MSE | MAE |
> | **ETTm2** | 96 | **0.172** | **0.256** | 0.174 | 0.255 | 0.174 | 0.255 |
> | | 192 | **0.236** | **0.298** | 0.243 | 0.302 | 0.239 | 0.299 |
> | | 336 | **0.300** | **0.338** | 0.304 | 0.341 | 0.301 | 0.340 |
> | | 720 | 0.406 | 0.400 | 0.399 | 0.397 | **0.395** | **0.396** |
> | **ECL** | 96 | **0.136** | **0.231** | 0.145 | 0.233 | 0.174 | 0.266 |
> | | 192 | **0.155** | **0.249** | 0.163 | 0.248 | 0.182 | 0.273 |
> | | 336 | **0.166** | **0.262** | 0.175 | 0.262 | 0.197 | 0.286 |
> | | 720 | **0.191** | **0.287** | 0.204 | 0.291 | 0.236 | 0.320 |
> | **weather** | 96 | **0.155** | **0.200** | 0.163 | 0.202 | 0.162 | 0.208 |
> | | 192 | **0.206** | **0.248** | 0.218 | 0.252 | 0.207 | 0.249 |
> | | 336 | **0.262** | **0.289** | 0.274 | 0.294 | 0.263 | 0.290 |
> | | 720 | 0.348 | 0.344 | 0.349 | 0.343 | **0.338** | **0.340** |
>
> We also provided a sensitivity analysis for the hyperparameters. We conducted experiments on multiple datasets for the most important parameters: the number of codebooks and the codebook size in A.2 Figure 6. The experiments show that these hyperparameters have a minimal impact on the model, demonstrating its robustness and generalization. For the Solar dataset, which is more complex than others due to factors like weather (e.g., cloud cover), increasing the number and size of the codebooks can improve performance. However, the improvement is not significant. Conversely, using more and larger codebooks will reduce the model's computational efficiency.

---

> ### Author Response · Authors · 2025-11-21
>
> We have added the results of VQ on the ECL dataset, as well as the results of SOFTS with VQ on the ETTm1, ETTm2, and ECL datasets. In most datasets, our method shows significant improvement, which proves the effectiveness of our approach.
>
> | Models || PatchTST || **PatchTST+VQ** || iTrans ||**iTrans+VQ** ||
> | :--- | :---: | :---: | :---: | :---: | :---: | :---: | :---: | :---: | :---: |
> | Metric || MSE | MAE | MSE | MAE | MSE | MAE | MSE | MAE |
> | ECL |96 | 0.164 | 0.251 | **0.154** | **0.245** | 0.148 | 0.240 | **0.138** | **0.235** |
> | |192 | 0.173 | 0.262 | **0.167** | **0.257** | 0.162 | **0.253** | **0.158** | 0.254 |
> | |336 | 0.190 | 0.279 | **0.184** | **0.274** | 0.178 | **0.269** | **0.173** | 0.271 |
> | |720 | 0.230 | 0.313 | **0.227** | **0.310** | 0.225 | 0.317 | **0.197** | **0.294** |
> || **Avg** | 0.189 | 0.276 | **0.183** | **0.271** | 0.178 | 0.270 | **0.166** | **0.263** |
>
> | Models || ECL || ETTm1 || ETTm2 ||
> | :--- | :--- | :---: | :---: | :---: | :---: | :---: | :---: |
> | Metric | | MSE | MAE | MSE | MAE | MSE | MAE |
> | **SOFTS** | 96 | 0.143 | 0.233 | 0.325 | 0.361 | 0.180 | 0.261 |
> | | 192 | 0.158 | 0.248 | 0.275 | 0.389 | 0.246 | 0.306 |
> | | 336 | 0.178 | 0.269 | 0.405 | 0.412 | 0.319 | 0.352 |
> | | 720 | 0.218 | 0.305 | **0.466** | **0.447** | **0.405** | **0.401** |
> | | **Avg** | 0.174 | 0.264 | 0.393 | 0.403 | 0.287 | 0.330 |
> | **SOFTS+VQ** | 96 | **0.136** | **0.229** | **0.319** | **0.356** | **0.175** | **0.257** |
> | | 192 | **0.155** | **0.249** | **0.365** | **0.382** | **0.243** | **0.301** |
> | | 336 | **0.170** | **0.267** | **0.399** | **0.405** | **0.312** | **0.345** |
> | | 720 | **0.208** | **0.301** | 0.477 | 0.450 | 0.408 | 0.403 |
> | | **Avg** | **0.167** | **0.261** | **0.390** | **0.398** | **0.284** | **0.326** |
>
> W4: We performed two ablation studies: (1) w/o-dr: we removed the codebooks from the discrete representation to use only the continuous representation, and (2) w/o-cr: we removed the continuous representation to use only the discrete representation. All other components remained unchanged. We observed a performance drop after removing either part, which confirms the rationale behind our model design. The impact on the overall model was small because the continuous embedding is not the core contribution of our method.
>
> | Models ||**Original** || w/o-dr || w/o-cr ||
> | :--- | :---: | :---: | :---: | :---: | :---: | :---: |:---: |
> | Metric || MSE | MAE | MSE | MAE | MSE | MAE |
> || ETTh1 | **0.445** | **0.439** | 0.450 | 0.441 | 0.446 | 0.441 |
> | |ETTh2 | **0.379** | **0.405** | 0.397 | 0.413 | 0.383 | 0.406 |
> | |ETTm1 | **0.385** | **0.395** | 0.394 | 0.402 | 0.386 | 0.396 |
> | |ETTm2 | **0.278** | **0.323** | 0.283 | 0.324 | 0.279 | 0.323 |

---

### Official Review · Reviewer_eEgj · 2025-10-31

**Soundness:** 3
**Presentation:** 3
**Contribution:** 3
**Rating:** 8
**Confidence:** 4

**Summary:**

The paper proposes the DCTS model to address limitations of traditional time series forecasting methods relying solely on continuous or discrete representations. Continuous methods are noisy and prone to numerical errors, while single-codebook discrete methods fail to cover full feature spaces. DCTS integrates a multi-codebook vector quantization-based Discrete Module for pattern extraction and a Continuous Module (with Fourier Transform and low-pass filtering) for noise reduction and information fusion. Experiments on 8 real datasets (e.g., ETTm1, Traffic) show it outperforms 8 baselines (e.g., iTransformer), with ablations confirming dual-module necessity and multi-codebooks’ superiority over single ones.

**Strengths:**

1. **Fuses dual representations for balanced performance**: DCTS combines continuous encoding’s expressive power (capturing fine-grained details) and discrete encoding’s pattern-abstracting ability (extracting key temporal patterns), avoiding pure continuous methods’ noise sensitivity and pure discrete methods’ detail loss
2. **Expands discrete space via multi-codebooks**: Unlike single-codebook vector quantization, it uses multiple independent codebooks to exponentially expand the encoding space (like combining "letters" into "words"), covering complex time series features without increasing codebook/vector size
3. **Reduces noise in continuous processing**: The Continuous Module uses Fourier Transform and low-pass filtering to remove high-frequency noise, focusing on critical trends and stabilizing predictions
4. **Validated effectiveness and generalizability**: It outperforms 8 baselines (e.g., iTransformer) on 8 real datasets (energy, traffic, etc.), and its Discrete Module can enhance other continuous models (e.g., 3.5% MSE gain for PatchTST)

**Weaknesses:**

1. DCTS relies on Euclidean distance for similarity measurement in vector quantization. When time series samples are phase-shifted, this distance metric tends to increase, leading to incorrect discrete code assignment and thus affecting the accuracy of discrete pattern extraction.
2. Although DTW is mentioned as a potential solution to handle phase-shifted samples by focusing on time series shape similarity, DCTS has not addressed the trade-off between DTW's effectiveness and its much lower computational efficiency compared to Euclidean distance.
3. While DCTS is validated on 8 real-world datasets, these datasets do not cover extremely complex time series scenarios (e.g., high-dimensional data with severe missing values or sudden abnormal fluctuations), leaving its adaptability in such scenarios unproven.

**Questions:**

see weaknesses.

---

> ### Author Response · Authors · 2025-11-21
>
> Thank you very much for your detailed review and valuable feedback on our work.
>
> We tested the performance of Euclidean distance and DTW (Dynamic Time Warping) distance for vector quantization using the same model parameters. Compared to Euclidean distance, DTW is difficult to parallelize, making its computational efficiency much lower. The DTW method requires several hours to train each epoch(17145s), while the method using Euclidean distance typically completes an epoch in just a few seconds. At the same time, the DTW method did not lead to better performance(0.377 to 0.376 MSE in ETTh1).
>
> We validated our model on the PEMS dataset. PEMS is a common dataset in the transportation field, which is more complex and has higher dimensions than other datasets. The results show that our method can maintain good performance in complex scenarios and on high-dimensional data.
> | Models || **Ours** || SOFTS || iTrans ||
> | :--- | :---: | :---: | :---: | :---: | :---: | :---: | :---: |
> | Metric || MSE | MAE | MSE | MAE | MSE | MAE |
> | **PEMS03** | 12 | **0.062** | **0.165** | 0.064 | 0.165 | 0.071 | 0.174 |
> | | 24 | **0.078** | **0.184** | 0.083 | 0.188 | 0.093 | 0.201 |
> | | 48 | **0.111** | **0.218** | 0.114 | 0.223 | 0.125 | 0.236 |
> | | 96 | 0.162 | 0.267 | **0.156** | **0.264** | 0.164 | 0.275 |
> | **PEMS04** | 12 | **0.071** | **0.175** | 0.074 | 0.176 | 0.078 | 0.183 |
> | | 24 | **0.080** | **0.184** | 0.088 | 0.194 | 0.095 | 0.205 |
> | | 48 | **0.100** | **0.208** | 0.110 | 0.219 | 0.120 | 0.233 |
> | | 96 | **0.133** | **0.240** | 0.135 | 0.244 | 0.150 | 0.262 |

---

### Author Response · Authors · 2025-12-03
**Summary**

Dear Area Chair and Reviewers,

For your convenience, we have summarized our approach and the key points discussed regarding our work.

This paper proposes DCTS, which integrates the discrete and continuous representations of time series. We obtain the discrete representation by applying Vector Quantization (VQ), which helps the model learn the key patterns of the time series. By integrating the discrete and continuous representations, our model can learn both the global patterns of the time series and retain the high expressive power of the continuous representation. This also prevents the model from overly focusing on fine details, thus enhancing its robustness. We adopted a multi-codebook independent encoding method, where each time series is represented by multiple independent codes, expanding the encoding space for stronger expressive ability.

**All reviewer concerns were addressed during the rebuttal phase:**
1. Extensive comparisons were conducted across multiple datasets against both reviewer-suggested and sota baselines. The proposed method demonstrates superior performance, achieving an average improvement of 3.0\% in MSE and 3.3\% in MAE. Additionally, experiments on the PEMS dataset were included to verify the adaptability of the method in complex scenarios.
2. New ablation studies are presented to validate the necessity of the codebooks and other components. Furthermore, DTW was evaluated as a quantization metric; results indicate that euclidean distance maintains both high performance and computational efficiency.
3. Comparative results with other VQ-based methods are provided, along with a discussion on the distinctiveness of the proposed approach relative to existing encoding methods. Moreover, results for multiple backbone models integrated with VQ across various datasets were included, demonstrating the effectiveness of incorporating discrete representations.
4. Hyperparameter analysis and prediction results for varying input lengths are included in the Appendix, demonstrating the robustness of the method. Codebook usage is also visualized to confirm the absence of dead codes.

**For Reviewer eEgj**
1. DTW was evaluated as the vector quantization method. It significantly reduced computational efficiency without more performance improvements (0.377 to 0.376 MSE in ETTh1).
2.  The model was validated on the PEMS dataset. Compared to the baselines, an average improvement of 7.8\% in MSE and 4.6\% in MAE was achieved, demonstrating the model's adaptability in complex scenarios.

**For Reviewer LGZf**
1. The distinctions between DCTS and existing methods regarding VQ and encoding method are discussed. Furthermore, the effectiveness of the proposed method is verified through new ablation studies.
2. VQ was integrated into new baselines and datasets. Across all experiments, an average improvement of 2.5\% in MSE and 1.6\% in MAE was achieved, demonstrating the effectiveness of the VQ method.
3. Comparative experiments against sota baselines were conducted on multiple datasets. The proposed method achieved an average improvement of 3.4\% in MSE and 2.1\% in MAE compared to the latest models.
4. A hyperparameter analysis of codebook quantity and size is provided, demonstrating the robustness of the model.

**For Reviewer mLa9**
1. Comparisons with the baselines suggested by the reviewer are provided. The proposed method shows an average improvement of 3.0\% in MSE and 4.0\% in MAE across multiple datasets compared to these new baselines.
2. A hyperparameter analysis of codebook quantity and size and prediction results for multiple input lengths were presented. These results demonstrate the model's robustness and its capability to handle long sequences.
3. The differences between the proposed VQ method and existing approaches are discussed. Additionally, new ablation studies are included to verify the significance of the codebooks.
4. Codebook usage frequency is visualized, confirming the absence of dead codes across multiple datasets.
5. The model was validated on the PEMS dataset. Compared to the baselines, an average improvement of 7.8\% in MSE and 4.6\% in MAE was achieved, demonstrating the model's adaptability in complex scenarios.

**For Reviewer bN1W**
1. Comparisons were made with sota baseline models. The proposed method achieved an average improvement of 3.2\% in MSE and 4.2\% in MAE across multiple datasets.
2.  Comparisons results for Sparse-VQ, which also employs VQ, are provided. The proposed method shows an improvement of 3.2\% in MSE and 1.8\% in MAE.
3. Prediction results for SDformer are also included. In multivariate long-term forecasting, the proposed method significantly outperforms the generative model SDformer.

We hope this brief record of our work and reviews will be helpful.

Sincerely,

The Authors

---

### Meta-Review · Area_Chair_W2s4 · 2026-01-09

**Summary:**

This paper proposes DCTS, a forecasting framework that fuses discrete and continuous representations of time-series data by combining a vector-quantized, multi-codebook discrete module with a continuous module based on frequency-domain filtering. The hybrid representation is used for downstream prediction and is shown to yield competitive performance against several established forecasting baselines on standard benchmarks.

Reviewers found the idea well motivated and the empirical evaluation solid, and the rebuttal significantly strengthened the paper by adding more recent baselines, additional datasets (including more complex traffic scenarios), and clearer ablations isolating the roles of the discrete and continuous components. However, concerns remain regarding the level of conceptual novelty and the differentiation from prior VQ-based time-series forecasting methods, as well as the overall magnitude of the empirical gains. While the revised experiments improve completeness and clarity, they do not yet fully address these concerns. As a result, the submission does not meet the bar for acceptance at this time.

**Reviewer Concerns:**

Reviewer eEgj: Raised concerns about Euclidean VQ under phase shifts and limited stress testing; rebuttal clarified DTW trade-offs and added complex PEMS results, largely addressing empirical scope.

Reviewer LGZf: Questioned novelty relative to RVQ/GRVQ and experimental completeness; rebuttal added baselines, sensitivity analyses, and clearer ablations, but novelty concerns persist.

Reviewer mLa9: Viewed the work as insufficiently novel with incomplete experiments; rebuttal substantially strengthened results via new baselines, hyperparameter/input-length analyses, and codebook visualizations.

Reviewer bN1W: Noted conservative baselines, missing discrete comparisons, and writing issues; rebuttal added stronger comparisons and ablations, improving support though presentation could still be polished.

**Reviewer Scores:**

Reviewer scores initially ranged from a clear accept to a strong reject (8/4/4/2), and while the rebuttal would likely shift one or even two borderline assessments slightly upward, remaining concerns about novelty and overall contribution suggest the paper would still sit near the boundary.

---

### Decision · Program_Chairs · 2026-01-26

Reject